# Multimodal Generative Models for Scalable Weakly-Supervised Learning

**Mike Wu**
Department of Computer Science
Stanford University
Stanford, CA 94025
wumike@stanford.edu

**Noah Goodman**
Departments of Computer Science and Psychology
Stanford University
Stanford, CA 94025
ngoodman@stanford.edu

## Abstract

Multiple modalities often co-occur when describing natural phenomena. Learning a joint representation of these modalities should yield deeper and more useful representations. Previous generative approaches to multi-modal input either do not learn a joint distribution or require additional computation to handle missing data. Here, we introduce a multimodal variational autoencoder (MVAE) that uses a product-of-experts inference network and a sub-sampled training paradigm to solve the multi-modal inference problem. Notably, our model shares parameters to efficiently learn under any combination of missing modalities. We apply the MVAE on four datasets and match state-of-the-art performance using many fewer parameters. In addition, we show that the MVAE is directly applicable to weakly-supervised learning, and is robust to incomplete supervision. We then consider two case studies, one of learning image transformations—edge detection, colorization, segmentation—as a set of modalities, followed by one of machine translation between two languages. We find appealing results across this range of tasks.

## 1 Introduction

Learning from diverse modalities has the potential to yield more generalizable representations. For instance, the visual appearance and tactile impression of an object converge on a more invariant abstract characterization [32]. Similarly, an image and a natural language caption can capture complimentary but converging information about a scene [28, 31]. While fully-supervised deep learning approaches can learn to bridge modalities, generative approaches promise to capture the joint distribution across modalities and flexibly support missing data. Indeed, multimodal data is *expensive* and *sparse*, leading to a *weakly supervised* setting of having only a small set of examples with all observations present, but having access to a larger dataset with one (or a subset of) modalities.

We propose a novel multimodal variational autoencoder (MVAE) to learn a joint distribution under weak supervision. The VAE [11] jointly trains a generative model, from latent variables to observations, with an *inference network* from observations to latents. Moving to multiple modalities and missing data, we would naively need an inference network for each combination of modalities. However, doing so would result in an exponential explosion in the number of trainable parameters. Assuming conditional independence among the modalities, we show that the correct inference network will be a product-of-experts [8], a structure which reduces the number of inference networks to one per modality. While the inference networks can be best trained separately, the generative model requires joint observations. Thus we propose a sub-sampled training paradigm in which fully-observed examples are treated as both fully and partially observed (for each gradient update). Altogether, this provides a novel and useful solution to the multi-modal inference problem.

We report experiments to measure the quality of the MVAE, comparing with previous models. We train on MNIST [14], binarized MNIST [13], MultiMNIST [6, 20], FashionMNIST [30], and CelebA [15]. Several of these datasets have complex modalities—character sequences, RGB images— requiring large inference networks with RNNs and CNNs. We show that the MVAE is able to support heavy encoders with thousands of parameters, matching state-of-the-art performance.

We then apply the MVAE to problems with more than two modalities. First, we revisit CelebA, this time fitting the model with each of the 18 attributes as an individual modality. Doing so, we find better performance from sharing of statistical strength. We further explore this question by choosing a handful of image transformations commonly studied in computer vision—colorization, edge detection, segmentation, etc.—and synthesizing a dataset by applying them to CelebA. We show that the MVAE can jointly learn these transformations by modeling them as modalities.

Finally, we investigate how the MVAE performs under incomplete supervision by reducing the number of multi-modal examples. We find that the MVAE is able to capture a good joint representation when only a small percentage of examples are multi-modal. To show real world applicability, we then investigate weak supervision on machine translation where each language is a modality.

## 2   Methods

A variational autoencoder (VAE) [11] is a latent variable generative model of the form $p_\theta(x, z) = p(z)p_\theta(x|z)$ where $p(z)$ is a prior, usually spherical Gaussian. The decoder, $p_\theta(x|z)$, consists of a deep neural net, with parameters $\theta$, composed with a simple likelihood (e.g. Bernoulli or Gaussian). The goal of training is to maximize the marginal likelihood of the data (the "evidence"); however since this is intractable, the evidence lower bound (ELBO) is instead optimized. The ELBO is defined via an inference network, $q_\phi(z|x)$, which serves as a tractable importance distribution:

$$\text{ELBO}(x) \triangleq \mathbb{E}_{q_\phi(z|x)}[\lambda \log p_\theta(x|z)] - \beta \, \text{KL}[q_\phi(z|x), p(z)] \tag{1}$$

where $\text{KL}[p, q]$ is the Kullback-Leibler divergence between distributions $p$ and $q$; $\beta$ [7] and $\lambda$ are weights balancing the terms in the ELBO. In practice, $\lambda = 1$ and $\beta$ is slowly annealed to 1 [2] to form a valid lower bound on the evidence. The ELBO is usually optimized (as we will do here) via stochastic gradient descent, using the reparameterization trick to estimate the gradient [11].

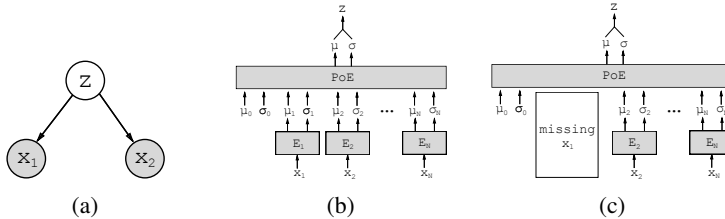

(a)   (b)   (c)

Figure 1: (a) Graphical model of the MVAE. Gray circles represent observed variables. (b) MVAE architecture with $N$ modalities. $E_i$ represents the $i$-th inference network; $\mu_i$ and $\sigma_i$ represent the $i$-th variational parameters; $\mu_0$ and $\sigma_0$ represent the prior parameters. The product-of-experts (PoE) combines all variational parameters in a principled and efficient manner. (c) If a modality is missing during training, we drop the respective inference network. Thus, the parameters of $E_1, ..., E_N$ are shared across different combinations of missing inputs.

In the multimodal setting we assume the $N$ modalities, $x_1, ..., x_N$, are conditionally independent given the common latent variable, $z$ (See Fig. 1a). That is we assume a generative model of the form $p_\theta(x_1, x_2, ..., x_N, z) = p(z)p_\theta(x_1|z)p_\theta(x_2|z)\cdots p_\theta(x_N|z)$. With this factorization, we can ignore unobserved modalities when evaluating the marginal likelihood. If we write a data point as the collection of modalities present, that is $X = \{x_i | i^{th} \text{ modality present}\}$, then the ELBO becomes:

$$\text{ELBO}(X) \triangleq \mathbb{E}_{q_\phi(z|X)}[\sum_{x_i \in X} \lambda_i \log p_\theta(x_i|z)] - \beta \, \text{KL}[q_\phi(z|X), p(z)]. \tag{2}$$

## 2.1 Approximating The Joint Posterior

The first obstacle to training the MVAE is specifying the $2^N$ inference networks, $q(z|X)$ for each subset of modalities $X \subseteq \{x_1, x_2, ..., x_N\}$. Previous work (e.g. [23, 26]) has assumed that the relationship between the joint- and single-modality inference networks is unpredictable (and therefore separate training is required). However, the optimal inference network $q(z|x_1, ..., x_N)$ would be the true posterior $p(z|x_1, ..., x_N)$. The conditional independence assumptions in the generative model imply a relation among joint- and single-modality posteriors:

$$
\begin{aligned}
p(z|x_1, ..., x_N) &= \frac{p(x_1, ..., x_N|z)p(z)}{p(x_1, ..., x_N)} = \frac{p(z)}{p(x_1, ..., x_N)} \prod_{i=1}^{N} p(x_i|z) \\
&= \frac{p(z)}{p(x_1, ..., x_N)} \prod_{i=1}^{N} \frac{p(z|x_i)p(x_i)}{p(z)} = \frac{\prod_{i=1}^{N} p(z|x_i)}{\prod_{i=1}^{N-1} p(z)} \cdot \frac{\prod_{i=1}^{N} p(x_i)}{p(x_1, ..., x_N)} \\
&\propto \frac{\prod_{i=1}^{N} p(z|x_i)}{\prod_{i=1}^{N-1} p(z)}
\end{aligned}
\tag{3}
$$

That is, the joint posterior is a product of individual posteriors, with an additional quotient by the prior. If we assume that the true posteriors for each individual factor $p(z|x_i)$ is properly contained in the family of its variational counterpart[1], $q(z|x_i)$, then Eqn. 3 suggests that the correct $q(z|x_1, ..., x_N)$ is a product and quotient of experts: $\frac{\prod_{i=1}^{N} q(z|x_i)}{\prod_{i=1}^{N-1} p(z)}$, which we call MVAE-Q.

Alternatively, if we approximate $p(z|x_i)$ with $q(z|x_i) \equiv \tilde{q}(z|x_i)p(z)$, where $\tilde{q}(z|x_i)$ is the underlying inference network, we can avoid the quotient term:

$$
p(z|x_1, ..., x_N) \propto \frac{\prod_{i=1}^{N} p(z|x_i)}{\prod_{i=1}^{N-1} p(z)} \approx \frac{\prod_{i=1}^{N}[\tilde{q}(z|x_i)p(z)]}{\prod_{i=1}^{N-1} p(z)} = p(z) \prod_{i=1}^{N} \tilde{q}(z|x_i).
\tag{4}
$$

In other words, we can use a product of experts (PoE), including a "prior expert", as the approximating distribution for the joint-posterior (Figure 1b). This representation is simpler and, as we describe below, numerically more stable. This derivation is easily extended to any subset of modalities yielding $q(z|X) \propto p(z) \prod_{x_i \in X} \tilde{q}(z|x_i)$ (Figure 1c). We refer to this version as MVAE.

The product and quotient distributions required above are not in general solvable in closed form. However, when $p(z)$ and $\tilde{q}(z|x_i)$ are Gaussian there is a simple analytical solution: a product of Gaussian experts is itself Gaussian [5] with mean $\mu = (\sum_i \mu_i T_i)(\sum_i T_i)^{-1}$ and covariance $V = (\sum_i T_i)^{-1}$, where $\mu_i$, $V_i$ are the parameters of the $i$-th Gaussian expert, and $T_i = V_i^{-1}$ is the inverse of the covariance. Similarly, given two Gaussian experts, $p_1(x)$ and $p_2(x)$, we can show that the quotient (QoE), $\frac{p_1(x)}{p_2(x)}$, is also a Gaussian with mean $\mu = (T_1\mu_1 - T_2\mu_2)(T_1 - T_2)^{-1}$ and covariance $V = (T_1 - T_2)^{-1}$, where $T_i = V_i^{-1}$. However, this distribution is well-defined only if $V_2 > V_1$ element-wise—a simple constraint that can be hard to deal with in practice. A full derivation for PoE and QoE can be found in the supplement.

Thus we can compute all $2^N$ multi-modal inference networks required for MVAE efficiently in terms of the $N$ uni-modal components, $\tilde{q}(z|x_i)$; the additional quotient needed by the MVAE-Q variant is also easily calculated but requires an added constraint on the variances.

## 2.2 Sub-sampled Training Paradigm

On the face of it, we can now train the MVAE by simply optimizing the evidence lower bound given in Eqn. 2. However, a product-of-Gaussians does not uniquely specify its component Gaussians. Hence, given a *complete* dataset, with no missing modalities, optimizing Eqn. 2 has an unfortunate consequence: we never train the individual inference networks (or small sub-networks) and thus do not know how to use them if presented with missing data at test time. Conversely, if we treat every observation as independent observations of each modality, we can adequately train the inference networks $\tilde{q}(z|x_i)$, but will fail to capture the relationship between modalities in the generative model.

We propose instead a simple training scheme that combines these extremes, including ELBO terms for whole and partial observations. For instance, with $N$ modalities, a complete example, $\{x_1, x_2, ..., x_N\}$ can be split into $2^N$ partial examples: $\{x_1\}, \{x_2, x_6\}, \{x_5, x_{N-4}, x_N\}, ....$ If we were to train using all $2^N$ subsets it would require evaluating $2^N$ ELBO terms. This is computationally intractable. To reduce the cost, we sub-sample which ELBO terms to optimize for every gradient step. Specifically, we choose (1) the ELBO using the product of all $N$ Gaussians, (2) all ELBO terms using a single modality, and (3) $k$ ELBO terms using $k$ randomly chosen subsets, $X_k$. For each minibatch, we thus evaluate a random subset of the $2^N$ ELBO terms. In expectation, we will be approximating the full objective. The sub-sampled objective can be written as:

$$\text{ELBO}(x_1, ..., x_N) + \sum_{i=1}^{N} \text{ELBO}(x_i) + \sum_{j=1}^{k} \text{ELBO}(X_j) \tag{5}$$

We explore the effect of $k$ in Sec. 5. A pleasant side-effect of this training scheme is that it generalizes to weakly-supervised learning. Given an example with missing data, $X = \{x_i | i^{th} \text{ modality present}\}$, we can still sample partial data from $X$, ignoring modalities that are missing.

## 3 Related Work

Given *two* modalities, $x_1$ and $x_2$, many variants of VAEs [11, 10] have been used to train generative models of the form $p(x_2|x_1)$, including conditional VAEs (CVAE) [21] and conditional multi-modal autoencoders (CMMA) [17]. Similar work has explored using hidden features from a VAE trained on images to generate captions, even in the weakly supervised setting [18]. Critically, these models are not bi-directional. We are more interested in studying models where we can condition interchangeably. For example, the BiVCCA [29] trains two VAEs together with interacting inference networks to facilitate two-way reconstruction. However, it does not attempt to directly model the joint distribution, which we find empirically to improve the ability of a model to learn the data distribution.

Several recent models have tried to capture the joint distribution explicitly. [23] introduced the joint multi-modal VAE (JMVAE), which learns $p(x_1, x_2)$ using a joint inference network, $q(z|x_1, x_2)$. To handle missing data at test time, the JMVAE collectively trains $q(z|x_1, x_2)$ with two other inference networks $q(z|x_1)$ and $q(z|x_2)$. The authors use an ELBO objective with two additional divergence terms to minimize the distance between the uni-modal and the multi-modal importance distributions. Unfortunately, the JMVAE trains a new inference network for each multi-modal subset, which we have previously argued in Sec. 2 to be intractable in the general setting.

Most recently, [26] introduce another objective for the bi-modal VAE, which they call the *triplet ELBO*. Like the MVAE, their model's joint inference network $q(z|x_1, x_2)$ combines variational distributions using a product-of-experts rule. Unlike the MVAE, the authors report a two-stage training process: using complete data, fit $q(z|x_1, x_2)$ and the decoders. Then, freezing $p(x_1|z)$ and $p(x_2|z)$, fit the uni-modal inference networks, $q(z|x_1)$ and $q(z|x_2)$ to handle missing data at test time. Crucially, because training is separated, the model has to fit 2 new inference networks to handle all combinations of missing data in stage two. While this paradigm is sufficient for two modalities, it does not generalize to the truly multi-modal case. To the best of our knowledge, the MVAE is the first deep generative model to explore more than two modalities efficiently. Moreover, the single-stage training of the MVAE makes it uniquely applicable to weakly-supervised learning.

Our proposed technique resembles established work in several ways. For example, PoE is reminiscent of a restricted Boltzmann machine (RBM), another latent variable model that has been applied to multi-modal learning [16, 22]. Like our inference networks, the RBM decomposes the posterior into a product of independent components. The benefit that a MVAE offers over a RBM is a simpler training algorithm via gradient descent rather than requiring contrastive divergence, yielding faster models that can handle more data. Our sub-sampling technique is somewhat similar to denoising [27, 16] where a subset of inputs are "partially destructed" to encourage robust representations in autoencoders. In our case, we can think of "robustness" as capturing the true marginal distributions.

## 4 Experiments

As in previous literature, we transform uni-modal datasets into multi-modal problems by treating labels as a second modality. We compare existing models (VAE, BiVCCA, JMVAE) to the MVAE

and show that we equal state-of-the-art performance on four image datasets: MNIST, FashionMNIST, MultiMNIST, and CelebA. For each dataset, we keep the network architectures consistent across models, varying only the objective and training procedure. Unless otherwise noted, given images $x_1$ and labels $x_2$, we set $\lambda_1 = 1$ and $\lambda_2 = 50$. We find that upweighting the reconstruction error for the low-dimensional modalities is important for learning a good joint distribution.

| Model | BinaryMNIST | MNIST | FashionMNIST | MultiMNIST | CelebA |
|---|---|---|---|---|---|
| VAE | 730240 | 730240 | 3409536 | 1316936 | 4070472 |
| CVAE | 735360 | 735360 | 3414656 | – | 4079688 |
| BiVCCA | 1063680 | 1063680 | 3742976 | 1841936 | 4447504 |
| JMVAE | 2061184 | 2061184 | 7682432 | 4075064 | 9052504 |
| MVAE-Q | 1063680 | 1063680 | 3742976 | 1841936 | 4447504 |
| MVAE | 1063680 | 1063680 | 3742976 | 1841936 | 4447504 |
| JMVAE19 | – | – | – | – | 3.6259e12 |
| MVAE19 | – | – | – | – | 10857048 |

Table 1: *Number of inference network parameters.* For a single dataset, each generative model uses the same inference network architecture(s) for each modality. Thus, the difference in parameters is solely due to how the inference networks interact in the model. We note that MVAE has the same number of parameters as BiVCCA. JMVAE19 and MVAE19 show the number of parameters using 19 inference networks when each of the attributes in CelebA is its own modality.

Our version of MultiMNIST contains between 0 and 4 digits composed together on a 50x50 canvas. Unlike [6], the digits are fixed in location. We generate the second modality by concatenating digits from top-left to bottom-right to form a string. As in literature, we use a RNN encoder and decoder [2]. Furthermore, we explore two versions of learning in CelebA, one where we treat the 18 attributes as a single modality, and one where we treat each attribute as its own modality for a total of 19. We denote the latter as MVAE19. In this scenario, to approximate the full objective, we set $k = 1$ for a total 21 ELBO terms (as in Eqn. 5). For complete details, including training hyperparameters and encoder/decoder architecture specification, refer to the supplement.

## 5 Evaluation

In the bi-modal setting with $x_1$ denoting the image and $x_2$ denoting the label, we measure the test marginal log-likelihood, $\log p(x_1)$, and test joint log-likelihood $\log p(x_1, x_2)$ using 100 importance samples in CelebA and 1000 samples in other datasets. In doing so, we have a choice of which inference network to use. For example, using $q(z|x_1)$, we estimate $\log p(x_1) \approx \log \mathbb{E}_{q(z|x_1)}[\frac{p(x_1|z)p(z)}{q(z|x_1)}]$. We also compute the test conditional log-likelihood $\log p(x_1|x_2)$, as a measure of classification performance, as done in [23]: $\log p(x_1|x_2) \approx \log \mathbb{E}_{q(z|x_2)}[\frac{p(x_1|z)p(x_2|z)p(z)}{q(z|x_2)}] - \log \mathbb{E}_{p(z)}[p(x_2|z)]$. In CelebA, we use 1000 samples to estimate $\mathbb{E}_{p(z)}[p(x_2|z)]$. In all others, we use 5000 samples. These marginal probabilities measure the ability of the model to capture the data distribution and its conditionals. Higher scoring models are better able to generate proper samples and convert between modalities, which is exactly what we find desirable in a generative model.

**Quality of the Inference Network** In all VAE-family models, the inference network functions as an importance distribution for approximating the intractable posterior. A better importance distribution, which more accurately approximates the posterior, results in importance weights with lower variance. Thus, we estimate the variance of the (log) importance weights as a measure of inference network quality (see Table 3).

Fig. 2 shows image samples and conditional image samples for each dataset using the image generative model. We find the samples to be good quality, and find conditional samples to be largely correctly matched to the target label. Table 2 shows test log-likelihoods for each model and dataset.[2] We see that MVAE performs on par with the state-of-the-art (JMVAE) while using far fewer parameters (see Table 1). When considering only $p(x_1)$ (i.e. the likelihood of the image modality alone), the

| Model | BinaryMNIST | MNIST | FashionMNIST | MultiMNIST | CelebA |
|---|---|---|---|---|---|
| | *Estimated log $p(x_1)$* | | | | |
| VAE | -86.313 | -91.126 | -232.758 | -152.835 | -6237.120 |
| BiVCCA | -87.354 | -92.089 | -233.634 | -202.490 | -7263.536 |
| JMVAE | -86.305 | -90.697 | -232.630 | -152.787 | -6237.967 |
| MVAE-Q | -91.665 | -96.028 | -236.081 | -166.580 | -6290.085 |
| MVAE | **-86.026** | **-90.619** | **-232.535** | **-152.761** | -6236.923 |
| MVAE19 | – | – | – | – | **-6236.109** |
| | *Estimated log $p(x_1, x_2)$* | | | | |
| JMVAE | -86.371 | **-90.769** | **-232.948** | **-153.101** | -6242.187 |
| MVAE-Q | -92.259 | -96.641 | -236.827 | -173.615 | -6294.861 |
| MVAE | **-86.255** | -90.859 | -233.007 | -153.469 | -6242.034 |
| MVAE19 | – | – | – | – | **-6239.944** |
| | *Estimated log $p(x_1\|x_2)$* | | | | |
| CVAE | **-83.448** | **-87.773** | **-229.667** | – | -6228.771 |
| JMVAE | -83.985 | -88.696 | -230.396 | **-145.977** | **-6231.468** |
| MVAE-Q | -90.024 | -94.347 | -234.514 | -163.302 | -6311.487 |
| MVAE | -83.970 | -88.569 | -230.695 | -147.027 | -6234.955 |
| MVAE19 | – | – | – | – | -6233.340 |

Table 2: Estimates (using $q(z|x_1)$) for marginal probabilities on the average test example. MVAE and JMVAE are roughly equivalent in data log-likelihood but as Table 1 shows, MVAE uses far fewer parameters. The CVAE is often better at capturing $p(x_1|x_2)$ but does not learn a joint distribution.

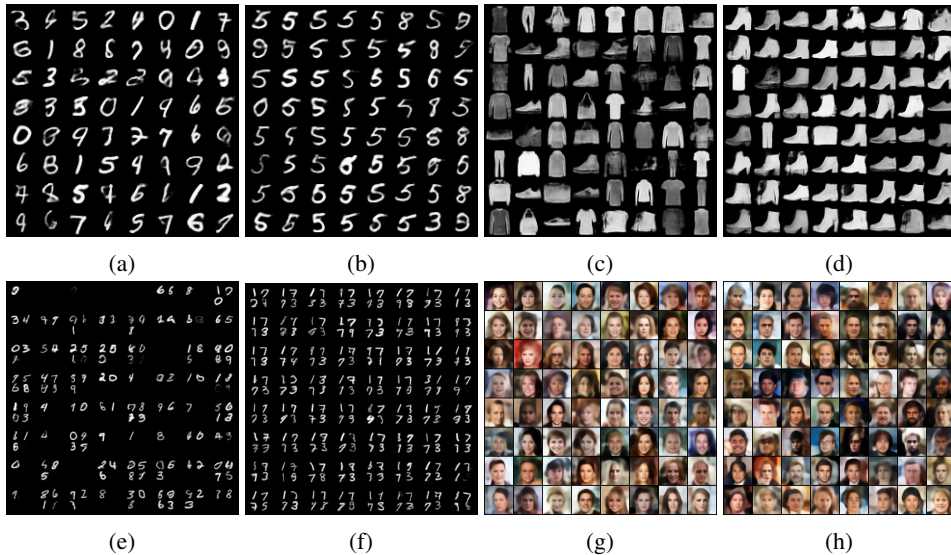

(a)  (b)  (c)  (d)

(e)  (f)  (g)  (h)

Figure 2: *Image samples using MVAE.* (a, c, e, g) show 64 images per dataset by sampling $z \sim p(z)$ and then generating via $p(x_1|z)$. Similarly, (b, d, f, h) show conditional image reconstructions by sampling $z \sim q(z|x_2)$ where (b) $x_2 = 5$, (d) $x_2 =$ Ankle boot, (f) $x_2 = 1773$, (h) $x_2 =$ Male.

MVAE also performs best, slightly beating even the image-only VAE, indicating that solving the harder multi-modal problem does not sacrifice any uni-modal model capacity and perhaps helps. On CelebA, MVAE19 (which treats features as independent modalities) out-performs the MVAE (which treats the feature vector as a single modality). This suggests that the PoE approach generalizes to a larger number of modalities, and that jointly training shares statistical strength. Moreover, we show in the supplement that the MVAE19 is robust to randomly dropping modalities.

Tables 3 show variances of log importance weights. The MVAE always produces lower variance than other methods that capture the joint distribution, and often lower than conditional or single-modality models. Furthermore, MVAE19 consistently produces lower variance than MVAE in CelebA. Overall, this suggests that the PoE approach used by the MVAE yields better inference networks.

| Model | BinaryMNIST | MNIST | FashionMNIST | MultiMNIST | CelebA |
|---|---|---|---|---|---|
| Variance of Marginal Log Importance Weights: $\mathrm{var}(\log(\frac{p(x_1,z)}{q(z\|x_1)}))$ | | | | | |
| VAE | 22.264 | 26.904 | 25.795 | 54.554 | **56.291** |
| BiVCCA | 55.846 | 93.885 | 33.930 | 185.709 | 429.045 |
| JMVAE | 39.427 | 37.479 | 53.697 | 84.186 | 331.865 |
| MVAE-Q | 34.300 | 37.463 | 34.285 | 69.099 | 100.072 |
| MVAE | **22.181** | **25.640** | **20.309** | **26.917** | 73.923 |
| MVAE19 | – | – | – | – | 71.640 |
| Variance of Joint Log Importance Weights: $\mathrm{var}(\log(\frac{p(x_1,x_2,z)}{q(z\|x_1)}))$ | | | | | |
| JMVAE | 41.003 | 40.126 | 56.640 | 91.850 | 334.887 |
| MVAE-Q | 34.615 | 38.190 | 34.908 | 64.556 | 101.238 |
| MVAE | **23.343** | **27.570** | **20.587** | **27.989** | 76.938 |
| MVAE19 | – | – | – | – | **72.030** |
| Variance of Conditional Log Importance Weights: $\mathrm{var}(\log(\frac{p(x_1,z\|x_2)}{q(z\|x_1)}))$ | | | | | |
| CVAE | 21.203 | **22.486** | **12.748** | – | **56.852** |
| JMVAE | 23.877 | 26.695 | 26.658 | 37.726 | 81.190 |
| MVAE-Q | 34.719 | 38.090 | 34.978 | 44.269 | 101.223 |
| MVAE | **19.478** | 25.899 | 18.443 | **16.822** | 73.885 |
| MVAE19 | – | – | – | – | 71.824 |

Table 3: Average variance of log importance weights for three marginal probabilities, estimated by importance sampling from $q(z|x_1)$. 1000 importance samples were used to approximate the variance. The lower the variance, the better quality the inference network.

**Effect of number of ELBO terms**   In the MVAE training paradigm, there is a hyperparameter $k$ that controls the number of sampled ELBO terms to approximate the intractable objective. To investigate its importance, we vary $k$ from 0 to 50 and for each, train a MVAE19 on CelebA. We find that increasing $k$ has little effect on data log-likelihood but reduces the variance of the importance distribution defined by the inference networks. In practice, we choose a small $k$ as a tradeoff between computation and a better importance distribution. See supplement for more details.

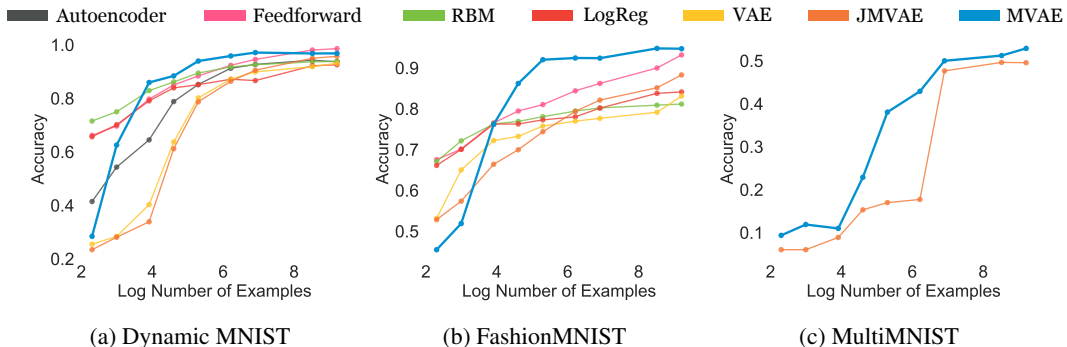

(a) Dynamic MNIST          (b) FashionMNIST          (c) MultiMNIST

Figure 3: *Effects of supervision level*. We plot the level of supervision as the log number of paired examples shown to each model. For MNIST and FashionMNIST, we predict the target class. For MultiMNIST, we predict the correct string representing each digit. We compare against a suite of baselines composed of models in relevant literature and commonly used classifiers. MVAE consistently beats all baselines in the *middle region* where there is both enough data to fit a deep model; in the fully-supervised regime, MVAE is competitive with feedforward deep networks. See supplement for accuracies.

## 5.1   Weakly Supervised Learning

For each dataset, we simulate incomplete supervision by randomly reserving a fraction of the dataset as multi-modal examples. The remaining data is split into two datasets: one with only the first modality, and one with only the second. These are shuffled to destroy any pairing. We examine the effect of supervision on the predictive task $p(x_2|x_1)$, e.g. predict the correct digit label, $x_2$, from an image $x_1$. For the MVAE, the total number of examples shown to the model is always fixed –

only the proportion of complete bi-modal examples is varied. We compare the performance of the MVAE against a suite of baseline models: (1) supervised neural network using the same architectures (with the stochastic layer removed) as in the MVAE; (2) logistic regression on raw pixels; (3) an autoencoder trained on the full set of images, followed by logistic regression on a subset of paired examples; we do something similar for (4) VAEs and (5) RBMs, where the internal latent state is used as input to the logistic regression; finally (6) we train the JMVAE ($\alpha = 0.01$ as suggested in [23]) on the subset of paired examples. Fig. 3 shows performance as we vary the level of supervision. For MultiMNIST, $x_2$ is a string (e.g. "6 8 1 2") representing the numbers in the image. We only include JMVAE as a baseline since it is not straightforward to output raw strings in a supervised manner.

We find that the MVAE surpasses all the baselines on a middle region when there are enough paired examples to sufficiently train the deep networks but not enough paired examples to learn a supervised network. This is especially emphasized in FashionMNIST, where the MVAE equals a fully supervised network even with two orders of magnitude less paired examples (see Fig. 3). Intuitively, these results suggest that the MVAE can effectively learn the joint distribution by bootstrapping from a larger set of uni-modal data. A second observation is that the MVAE almost always performs better than the JMVAE. This discrepancy is likely due to directly optimizing the marginal distributions rather than minimizing distance between several variational posteriors. We noticed empirically that in the JMVAE, using the samples from $q(z|x, y)$ did much better (in accuracy) than samples from $q(z|x)$.

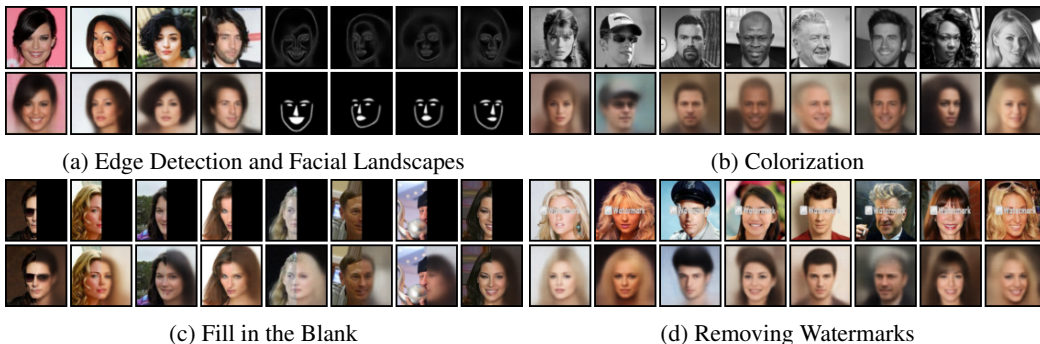

| (a) Edge Detection and Facial Landscapes | (b) Colorization |
| --- | --- |
| (c) Fill in the Blank | (d) Removing Watermarks |

Figure 4: *Learning Computer Vision Transformations:* (a) 4 ground truth images randomly chosen from CelebA along with reconstructed images, edges, and facial landscape masks; (b) reconstructed color images; (c) image completion via reconstruction; (d) reconstructed images with the watermark removed. See supplement for a larger version with more samples.

## 6 Case study: Computer Vision Applications

We use the MVAE to learn image transformations (and their inverses) as conditional distributions. In particular, we focus on colorization, edge detection, facial landmark segmentation, image completion, and watermark removal. The original image is itself a modality, for a total of six.

To build the dataset, we apply ground-truth transformations to CelebA. For *colorization*, we transform RGB colors to grayscale. For *image completion*, half of the image is replaced with black pixels. For *watermark removal*, we overlay a generic watermark. To extract edges, we use the Canny detector [4] from Scikit-Image [24]. To compute facial landscape masks, we use dlib [9] and OpenCV [3].

We fit a MVAE with 250 latent dimensions and $k{=}1$. We use Adam with a $10^{-4}$ learning rate, a batch size of 50, $\lambda_i = 1$ for $i = 1, ..., N$, $\beta$ annealing for 20 out of 100 epochs. Fig. 4 shows samples showcasing different learned transformations. In Fig. 4a we encode the original image with the learned encoder, then decode the transformed image with the learned generative model. We see reasonable reconstruction, and good facial landscape and edge extraction. In Figs.4b, 4c, 4d we go in the opposite direction, encoding a transformed image and then sampling from the generative model to reconstruct the original. The results are again quite good: reconstructed half-images agree on gaze direction and hair color, colorizations are reasonable, and all trace of the watermark is removed. (Though the reconstructed images still suffer from the same blurriness that VAEs do [33].)

# 7 Case study: Machine Translation

As a second case study we explore machine translation with weak supervision – that is, where only a small subset of data consist of translated sentence pairs. Many of the popular translation models [25] are fully supervised with millions of parameters and trained on datasets with tens of millions of paired examples. Yet aligning text across languages is very costly, requiring input from expert human translators. Even the unsupervised machine translation literature relies on large bilingual dictionaries, strong pre-trained language models, or synthetic datasets [12, 1, 19]. These factors make weak supervision particularly intriguing.

We use the English-Vietnamese dataset (113K sentence pairs) from IWSLT 2015 and treat English (en)

| Num. Aligned Data (%) | Test log $p(x)$ |
|---|---|
| 133 (0.1%) | $-558.88 \pm 3.56$ |
| 665 (0.5%) | $-494.76 \pm 4.18$ |
| 1330 (1%) | $-483.23 \pm 5.81$ |
| 6650 (5%) | $-478.75 \pm 3.00$ |
| 13300 (10%) | $-478.04 \pm 4.95$ |
| 133000 (100%) | $-478.12 \pm 3.02$ |

Table 4: *Weakly supervised translation.* Log likelihoods on a test set, averaged over 3 runs. Notably, we find good performance with a small fraction of paired examples.

and Vietnamese (vi) as two modalities. We train the MVAE with 100 latent dimensions for 100 epochs ($\lambda_{en} = \lambda_{vi} = 1$). We use the RNN architectures from [2] with a maximum sequence length of 70 tokens. As in [2], word dropout and KL annealing are crucial to prevent latent collapse.

| Type | Sentence |
|---|---|
| $x_{en} \sim p_{data}$ | this was one of the highest points in my life. |
| $x_{vi} \sim p(x_{vi}\|z(x_{en}))$ | Đó là một gian tôi vời của cuộc đời tôi. |
| GOOGLE($x_{vi}$) | It was a great time of my life. |
| $x_{en} \sim p_{data}$ | the project's also made a big difference in the lives of the people . |
| $x_{vi} \sim p(x_{vi}\|z(x_{en}))$ | tôi án này được ra một Điều lớn lao cuộc sống của chúng người sống chữa hưởng . |
| GOOGLE($x_{vi}$) | this project is a great thing for the lives of people who live and thrive . |
| $x_{vi} \sim p_{data}$ | trước tiên , tại sao chúng lại có ấn tượng xấu như vậy ? |
| $x_{en} \sim p(x_{en}\|z(x_{vi}))$ | first of all, you do not a good job ? |
| GOOGLE($x_{vi}$) | First, why are they so bad? |
| $x_{vi} \sim p_{data}$ | Ông ngoại của tôi là một người thật đáng <unk> phục vào thời ấy . |
| $x_{en} \sim p(x_{en}\|z(x_{vi}))$ | grandfather is the best experience of me family . |
| GOOGLE($x_{vi}$) | My grandfather was a worthy person at the time . |

Table 5: Examples of (1) translating English to Vietnamese by sampling from $p(x_{vi}|z)$ where $z \sim q(z|x_{en})$, and (2) the inverse. We use Google Translate (GOOGLE) for ground-truth.

With only 1% of aligned examples, the MVAE is able to describe test data almost as well as it could with a fully supervised dataset (Table 4). With 5% aligned examples, the model reaches maximum performance. Table 5 shows examples of translation forwards and backwards between English and Vietnamese. See supplement for more examples. We find that many of the translations are not extremely faithful but interestingly capture a close interpretation to the true meaning. While these results are not competitive to state-of-the-art translation, they are remarkable given the very weak supervision. Future work should investigate combining MVAE with modern translation architectures (e.g. transformers, attention).

# 8 Conclusion

We introduced a multi-modal variational autoencoder with a new training paradigm that learns a joint distribution and is robust to missing data. By optimizing the ELBO with multi-modal and uni-modal examples, we fully utilize the product-of-experts structure to share inference network parameters in a fashion that scales to an arbitrary number of modalities. We find that the MVAE matches the state-of-the-art on four bi-modal datasets, and shows promise on two real world datasets.

**Acknowledgments**

MW is supported by NSF GRFP and the Google Cloud Platform Education grant. NDG is supported under DARPA PPAML through the U.S. AFRL under Cooperative Agreement FA8750-14-2-0006. We thank Robert X. D. Hawkins and Ben Peloquin for helpful discussions.

## Footnotes

[1]Without this assumption, the best approximation to a product of factors may not be the product of the best approximations for each individual factor. But, the product of $q(z|x_i)$ is still a tractable family of approximations.

[2]These results used $q(z|x_1)$ as the importance distribution. See supplement for similar results using $q(z|x_1, x_2)$. Because importance sampling with either $q(z|x_1)$ or $q(z|x_1, x_2)$ yields an unbiased estimator of marginal likelihood, we expect the log-likelihoods to agree asymptotically.

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
