[Supplementary Material · wu_2018_supplement.pdf]

# Supplementary Material

**Mike Wu**
Department of Computer Science
Stanford University
Stanford, CA 94025
`wumike@stanford.edu`

**Noah Goodman**
Department of Computer Science and Psychology
Stanford University
Stanford, CA 94025
`ngoodman@stanford.edu`

## 1  Dataset Descriptions

**MNIST/BinaryMNIST**   We use the MNIST hand-written digits dataset [6] with 50,000 examples for training, 10,000 validation, 10,000 testing. We also train on a binarized version to align with previous work [5]. As in [8], we use the Adam optimizer [4] with a learning rate of 1e-3, a minibatch size of 100, 64 latent dimensions, and train for 500 epochs. We anneal $\beta$ from 0 to 1 linearly for the first 200 epochs. For the encoders and decoders, we use MLPs with 2 hidden layers of 512 nodes. We model $p(x_1|z)$ with a Bernoulli likelihood and $p(x_2|z)$ with a multinomial likelihood.

**FashionMNIST**   This is an MNIST-like fashion dataset containing 28 x 28 grayscale images of clothing from 10 classes—skirts, shoes, t-shirts, etc [9]. We use identical hyperparameters as in MNIST. However, we employ a miniature DCGAN [7] for the image encoder and decoder.

**MultiMNIST**   This is variant of MNIST where between 0 and 4 digits are composed together on a 50x50 canvas. Unlike [3], the digits are fixed in location. We generate the text modality by concatenating the digit classes from top-left to bottom-right. We use 100 latent dimensions, with the remaining hyperparameters as in MNIST. For the image encoder and decoder, we retool the DCGAN architecture from [7]. For the text encoder, we use a bidirectional GRU with 200 hidden units. For the text decoder, we first define a vocabulary with ten digits, a start token, and stop token. Provided a start token, we feed it through a 2-layer GRU, linear layers, and a softmax. We sample a new character and repeat until generating a stop token. We note that previous work has not explored RNN-VAE inference networks in multi-modal learning, which we show to work well with the MVAE.

**CelebA**   The CelebFaces and Attributes (CelebA) dataset [10] contains over 200k images of celebrities. Each image is tagged with 40 attributes i.e. wears glasses, or has bangs. We use the aligned and cropped version with a selected 18 visually distinctive attributes, as done in [**?** ]. Images are rescaled to 64x64. For the first experiment, we treat images as one modality, $x_1$, and attributes as a second modality, $x_2$ where a single inference network predicts all 18 attributes. We also explore a variation of MVAE, called MVAE19, where we treat each attribute as its own modality for a total of 19. To approximate the full objective, we set $k = 1$ for a total 21 ELBO terms. We use Adam with a learning rate of $10^{-4}$, a minibatch size of 100, and anneal KL for the first 20 of 100 epochs. We again use DCGAN for image networks. For the attribute encoder and decoder, we use an MLP with 2 hidden layers of size 512. For MVAE19, we have 18 such encoders and decoders.

## 2  Product of a Finite Number of Gaussians

In this section, we provide the derivation for the parameters of a product of Gaussian experts (PoE). Derivation is summarized from [1].

**Lemma 2.1.** *Give a finite number $N$ of multi-dimensional Gaussian distributions $p_i(x)$ with mean $\mu_i$, covariance $V_i$ for $i = 1, ..., N$, the product $\prod_{i=1}^{N} p_i(x)$ is itself Gaussian with mean $(\sum_{i=1}^{N} T_i \mu_i)(\sum_{i=1}^{N} T_i)^{-1}$ and covariance $(\sum_{i=1}^{N} T_i)^{-1}$ where $T_i = V_i^{-1}$.*

*Proof.* We write the probability density of a Gaussian distribution in canonical form as $K \exp\{\eta^T x - \frac{1}{2} x^T \Lambda x\}$ where $K$ is a normalizing constant, $\Lambda = V^{-1}$, $\eta = V^{-1}\mu$. We then write the product of $N$ Gaussians distributions $\prod_{i=1}^{N} p_i \propto \exp\{(\sum_{i=1}^{N} \eta_i)^T x - \frac{1}{2} x^T (\sum_{i=1}^{N} \Lambda_i) x\}$. We note that this product itself has the form of a Gaussian distribution with $\eta = \sum_{i=1}^{N} \eta_i$ and $\Lambda = \sum_{i=1}^{N} \Lambda_i$. Converting back from canonical form, we see that the product Gaussian has mean $\mu = (\sum_{i=1}^{N} T_i \mu_i)(\sum_{i=1}^{N} T_i)^{-1}$ and covariance $V = (\sum_{i=1}^{N} T_i)^{-1}$. $\qquad\square$

## 3 Quotient of Two Gaussians

Similarly, we may derive the form of a quotient of two Gaussian distributions (QoE).

**Lemma 3.1.** *Give two multi-dimensional Gaussian distributions $p(x)$ and $q(x)$ with mean $\mu_p$ and $\mu_q$, and covariance $V_p$ and $V_q$ respectively, the quotient $\frac{p(x)}{q(x)}$ is itself Gaussian with mean $(T_p \mu_p - T_q \mu_q)(T_p - T_q)^{-1}$ and covariance $(T_p - T_q)^{-1}$ where $T_i(x) = V_i^{-1}(x)$.*

*Proof.* We again write the probability density of a Gaussian distribution as $K \exp\{\eta^T x - \frac{1}{2} x^T \Lambda x\}$. We then write the quotient of two Gaussians $p$ and $q$ as $\frac{K_p \exp\{\eta_p^T x - \frac{1}{2} x^T \Lambda_p x\}}{K_q \exp\{\eta_q^T x - \frac{1}{2} x^T \Lambda_q x\}} \propto \exp\{(\eta_p - \eta_q)^T x - \frac{1}{2} x^T (\Lambda_p - \Lambda_q) x\}$. This defines a new Gaussian distribution with $\Lambda = V_p^{-1} - V_q^{-1}$ and $\eta = V_p^{-1}\mu_p - V_q^{-1}\mu_q$. If we let $T_p = V_p^{-1}$ and $T_q = V_q^{-1}$, then we see that $V = \Lambda^{-1} = (T_p - T_q)^{-1}$ and $\mu = \eta V^{-1} = (T_p \mu_p - T_q \mu_q)(T_p - T_q)^{-1}$. $\qquad\square$

The QoE suggests that the constraint $T_p > T_q \Rightarrow V_p^{-1} > V_q^{-1}$ must hold for the resulting Gaussian to be well-defined. In our experiments, $p$ is usually a product of Gaussians, and $q$ is a product of prior Gaussians (see Eqn 3 in main paper). Given $N$ modalities, we can decompose $V_p = \sum_{i=1}^{N} V_i^{-1}$ and $V_q = \sum_{i=1}^{N-1} 1 = N - 1$ where the prior is a unit Gaussian with variance 1. Thus, the constraint can be rewritten as $\sum_{i=1}^{N} V_i^{-1} > N - 1$, which is satisfied if $V_i > \frac{N}{N-1}, i = 1, ..., N$. One benefit of using the regularized importance distribution $q(z|x)p(z)$ is to remove the need for this constraint. To fit MVAE without a universal expert, we add an additional nonlinearity to each inference network such that the variance is fed into a rescaled sigmoid: $V = \frac{N}{N-1} \cdot \text{sigm}(V)$.

## 4 Additional Results using the Joint Inference Network

In the main paper, we reported marginal probabilities using $q(z|x_1)$ and showed that MVAE is state-of-the-art. Here we similarly compute marginal probabilities but using $q(z|x_1, x_2)$. Because importance sampling with either induced distribution yields an unbiased estimate, using a large number of samples should result in very similar log-likelihoods. Indeed, we find that the results do not differ much from the main paper: MVAE is still at the state-at-the-art.

## 5 Model Architectures

Here we specify the design of inference networks and decoders used for each dataset.

## 6 More on Weak Supervision

In the main paper, we showed that we do not need that many complete examples to learn a good joint distribution with two modalities. Here, we explore the robustness of our model with missing data under more modalities. Using MVAE19 (19 modalities) on CelebA, we can conduct a different weak supervision experiment: given a complete multi-modal example $(x_1, ..., x_{19})$, randomly keep $x_i$ with

| Model | BinaryMNIST | MNIST | FashionMNIST | MultiMNIST | CelebA |
|---|---|---|---|---|---|
| | | Estimating $\log p(x_1)$ using $q(z|x_1, x_2)$ | | | |
| JMVAE | -86.234 | -90.962 | **-232.401** | -153.026 | **-6234.542** |
| MVAE | **-86.051** | **-90.616** | -232.539 | **-152.826** | -6237.104 |
| MVAE19 | – | – | – | – | -6236.113 |
| | | Estimating $\log p(x_1, x_2)$ using $q(z|x_1, x_2)$ | | | |
| JMVAE | -86.304 | -91.031 | **-232.700** | **-153.320** | **-6238.280** |
| MVAE | **-86.278** | **-90.851** | -233.007 | -153.478 | -6241.621 |
| MVAE19 | – | – | – | – | -6239.957 |
| | | Estimating $\log p(x_1|x_2)$ using $q(z|x_1, x_2)$ | | | |
| JMVAE | **-83.820** | **-88.436** | **-230.651** | **-145.761** | -6235.330 |
| MVAE | -83.940 | -88.558 | -230.699 | -147.009 | -6235.368 |
| MVAE19 | – | – | – | – | **-6233.330** |

Table 1: Similar estimates as in Table 2 (in main paper) but using $q(z|x_1, x_2)$ as an importance distribution (instead of $q(z|x_1)$). Because VAE and CVAE do not have a multi-modal inference network, they are excluded. Again, we show that the MVAE is able to match state-of-the-art.

| Model | BinaryMNIST | MNIST | FashionMNIST | MultiMNIST | CelebA |
|---|---|---|---|---|---|
| | | Variance of Marginal Log Importance Weights: $\mathrm{var}(\log(\frac{p(x_1, z)}{q(z|x_1, x_2)}))$ | | | |
| JMVAE | 22.387 | **24.962** | 28.443 | 35.822 | 80.808 |
| MVAE | **21.791** | 25.741 | **18.092** | **16.437** | 73.871 |
| MVAE19 | – | – | – | – | **71.546** |
| | | Variance of Joint Log Importance Weights: $\mathrm{var}(\log(\frac{p(x_1, x_2, z)}{q(z|x_1, x_2)}))$ | | | |
| JMVAE | 23.309 | 26.767 | 29.874 | 38.298 | 81.312 |
| MVAE | **21.917** | **26.057** | **18.263** | **16.672** | 74.968 |
| MVAE19 | – | – | – | – | **71.953** |
| | | Variance of Conditional Log Importance Weights: $\mathrm{var}(\log(\frac{p(x_1, z|x_2)}{q(z|x_1, x_2)}))$ | | | |
| JMVAE | 40.646 | 40.086 | 56.452 | 92.683 | 335.046 |
| MVAE | **23.035** | **27.652** | **19.934** | **28.649** | 77.516 |
| MVAE19 | – | – | – | – | **71.603** |

Table 2: Average variance of log importance weights for marginal, joint, and conditional distributions using $q(z|x_1, x_2)$. Lower variances suggest better inference networks.

Figure 1: *MVAE architectures on MNIST:* (a) $q(z|x_1)$, (b) $p(x_1|z)$, (c) $q(z|x_2)$, (d) $q(x_2|z)$ where $x_1$ specifies an image and $x_2$ specifies a digit label.

probability $p$ for each $i = 1, ..., 19$. Doing so for all examples in the training set, we simulate the effect of missing modalities beyond the bi-modal setting. Here, the number of examples shown to the model is dependent on $p$ e.g. $p = 0.5$ suggests that on average, 1 out of every 2 $x_i$ are dropped. We vary $p$ from 0.001 to 1, train from scratch, and plot (1) the prediction accuracy per attribute and (2) the various data log-likelihoods. From Figure 5, we conclude that the method is fairly robust to missing data. Even with $p = 0.1$, we still see accuracy close to the prediction accuracy with full data.

Figure 2: *MVAE architectures on FashionMNIST:* (a) $q(z|x_1)$, (b) $p(x_1|z)$, (c) $q(z|x_2)$, (d) $q(x_2|z)$ where $x_1$ specifies an image and $x_2$ specifies a clothing label.

Figure 3: *MVAE architectures on MultiMNIST:* (a) $q(z|x_1)$, (b) $p(x_1|z)$, (c) $q(z|x_2)$, (d) $q(x_2|z)$ where $x_1$ specifies an image and $x_2$ specifies a string of 4 digits.

# 7 Table of Weak Supervision Results

In the paper, we showed a series of plots detailing the performance the MVAE among many baselines on a weak supervision task. Here we provide tables detailing other numbers.

| Model | 0.1% | 0.2% | 0.5% | 1% | 2% | 5% | 10% | 50% | 100% |
|---|---|---|---|---|---|---|---|---|---|
| AE | 0.4143 | 0.5429 | 0.6448 | 0.788 | 0.8519 | 0.9124 | 0.9269 | 0.9423 | 0.9369 |
| NN | 0.6618 | 0.6964 | 0.7971 | 0.8499 | 0.8838 | 0.9235 | 0.9455 | 0.9806 | 0.9857 |
| LOGREG | 0.6565 | 0.7014 | 0.7907 | 0.8391 | 0.8510 | 0.8713 | 0.8665 | 0.9217 | 0.9255 |
| RBM | 0.7152 | 0.7496 | 0.8288 | 0.8614 | 0.8946 | 0.917 | 0.9257 | 0.9365 | 0.9379 |
| VAE | 0.2547 | 0.284 | 0.4026 | 0.6369 | 0.8016 | 0.8717 | 0.8989 | 0.9183 | 0.9311 |
| JMVAE | 0.2342 | 0.2809 | 0.3386 | 0.6116 | 0.7869 | 0.8638 | 0.9051 | 0.9498 | 0.9572 |
| MVAE | 0.2842 | 0.6254 | 0.8593 | 0.8838 | 0.9394 | 0.9584 | 0.9711 | 0.9678 | 0.9681 |

Table 3: Performance of several models on MNIST with a fraction of paired examples. Here we compute the accuracy (out of 1) of predicting the correct digit in each image.

Figure 4: *MVAE architectures on CelebA:* (a) $q(z|x_1)$, (b) $p(x_1|z)$, (c) $q(z|x_2)$, (d) $q(x_2|z)$ where $x_1$ specifies an image and $x_2$ specifies a 18 attributes.

(a) $p(y|x)$  (b) $\log p(x,y)$  (c) $\log p(x)$  (d) $\log p(x|y)$

Figure 5: We randomly drop input features $x_i$ with probability $p$. Figure (a) shows the effect of increasing $p$ from 0.001 to 1 on the accuracy of sampling the correct attribute given an image. Figure (b) and (c) show changes in log marginal and log conditional approximations as $p$ increases. In all cases, we see close-to-best performance using only 10% of the complete data.

# 8 Details on Weak Supervision Baselines

The VAE used the same image encoder as the MVAE. JMVAE used identical architectures as the MVAE with a hyperparameter $\alpha = 0.01$. The RBM has a single layer with 128 hidden nodes and is trained using contrastive divergence. NN uses the image encoder and label/string decoder as in MVAE, thereby being a fair comparison to supervised learning. For MNIST, we trained each model for 500 epochs. For FashionMNIST and MultiMNIST, we trained each model for 100 epochs. All other hyperparameters were kept constant between models.

| Model | 0.1% | 0.2% | 0.5% | 1% | 2% | 5% | 10% | 50% | 100% |
|---|---|---|---|---|---|---|---|---|---|
| NN | 0.6755 | 0.701 | 0.7654 | 0.7944 | 0.8102 | 0.8439 | 0.862 | 0.8998 | 0.9318 |
| LOGREG | 0.6612 | 0.7005 | 0.7624 | 0.7627 | 0.7728 | 0.7802 | 0.8015 | 0.8377 | 0.8412 |
| RBM | 0.6708 | 0.7214 | 0.7628 | 0.7690 | 0.7805 | 0.7943 | 0.8021 | 0.8088 | 0.8115 |
| VAE | 0.5316 | 0.6502 | 0.7221 | 0.7324 | 0.7576 | 0.7697 | 0.7765 | 0.7914 | 0.8311 |
| JMVAE | 0.5284 | 0.5737 | 0.6641 | 0.6996 | 0.7437 | 0.7937 | 0.8212 | 0.8514 | 0.8828 |
| MVAE | 0.4548 | 0.5189 | 0.7619 | 0.8619 | 0.9201 | 0.9243 | 0.9239 | 0.9478 | 0.947 |

Table 4: Performance of several models on FashionMNIST with a fraction of paired examples. Here we compute the accuracy of predicting the correct class of attire in each image.

| Model | 0.1% | 0.2% | 0.5% | 1% | 2% | 5% | 10% | 50% | 100% |
|---|---|---|---|---|---|---|---|---|---|
| JMVAE | 0.0603 | 0.0603 | 0.0888 | 0.1531 | 0.1699 | 0.1772 | 0.4765 | 0.4962 | 0.4955 |
| MVAE | 0.09363 | 0.1189 | 0.1098 | 0.2287 | 0.3805 | 0.4289 | 0.4999 | 0.5121 | 0.5288 |

Table 5: Performance of several models on MultiMNIST with a fraction of paired examples. Here compute the average accuracy of predicting each digit correct (by decomposing the string into individual digits, at most 4).

## 9 More of the effects of sampling more ELBO terms

In the main paper, we stated that with higher $k$ (sampling more ELBO terms), we see a steady decrease in variance. This drop in variance can be attributed to two factors: (1) additional un-correlated randomness from sampling more when reparametrizing for each ELBO [2], or (2) additional ELBO terms to better approximate the intractable objective. Fig. 6 (c) shows that the variance still drops consistently when using a fixed $\epsilon \sim N(0,1)$ for computing all ELBO terms, indicating independent contributions of additional ELBO terms and additional randomness.

Figure 6: Effect of approximating the MVAE objective with more ELBO terms on (a) the joint log-likelihood and (b) the variance of the log importance weights over 3 independent runs. Similarly, (c) compute the variance but fixes a single $\epsilon \sim N(0,1)$ when reparametrizing for each ELBO. (b) and (c) imply that switching from $k = 0$ to $k = 1$ greatly reduces the variance in the importance distribution defined by the inference network(s).

## 10 More on the Computer Vision Transformations

We copy Fig. 4 in the main paper but show more samples and increase the size of each image for visibility. The MVAE is able to learn all 6 transformations jointly under the PoE inference network.

## 11 More on Machine Translation

We provide more samples on (1) sampling joint (English, Vietnamese) pairs of sentences from the prior $N(0,1)$, (2) translating English to Vietnamese by sampling from $p(x_{en}|z)$ where $z \sim q(z|x_{vi})$, and (3) translating Vietnamese to English by sampling from $p(x_{vi}|z)$ where $z \sim q(z|x_{en})$. Refer to the main for analysis and explanation.

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

Figure 10: *Removing Watermarks*: The top row shows ground truth CelebA images, each with an added watermark. The bottom row shows the reconstructed image with the watermark removed.

| Type | Sentence |
|---|---|
| $x_{en} \sim p(x_{en}\|z = z_0)$ | it's a problem . |
| $x_{vi} \sim p(x_{vi}\|z = z_0)$ | nó là một công việc . |
| GOOGLETRANSLATE$(x_{vi})$ | it is a job . |
| $x_{en} \sim p(x_{en}\|z = z_0)$ | we have an idea . |
| $x_{vi} \sim p(x_{vi}\|z = z_0)$ | chúng tôi có thể làm được . |
| GOOGLETRANSLATE$(x_{vi})$ | we can do it . |
| $x_{en} \sim p(x_{en}\|z = z_0)$ | And as you can see , this is a very powerful effect of word of mouth . |
| $x_{vi} \sim p(x_{vi}\|z = z_0)$ | và một trong những điều này đã xảy ra với những người khác , và chúng tôi đã có một số người trong số các bạn đã từng nghe về những điều này . |
| GOOGLETRANSLATE$(x_{vi})$ | and one of these has happened to other people, and we've had some of you guys already heard about this . |
| $x_{en} \sim p(x_{en}\|z = z_0)$ | this is a photograph of my life . |
| $x_{vi} \sim p(x_{vi}\|z = z_0)$ | Đây là một bức ảnh . |
| GOOGLETRANSLATE$(x_{vi})$ | this is a photo . |
| $x_{en} \sim p(x_{en}\|z = z_0)$ | thank you . |
| $x_{vi} \sim p(x_{vi}\|z = z_0)$ | xin cảm ơn . |
| GOOGLETRANSLATE$(x_{vi})$ | thank you . |
| $x_{en} \sim p(x_{en}\|z = z_0)$ | i'm not kidding . |
| $x_{vi} \sim p(x_{vi}\|z = z_0)$ | tôi không nói đùa . |
| GOOGLETRANSLATE$(x_{vi})$ | i am not joking . |

Table 6: A few examples of "paired" reconstructions from a single sample $z_0 \sim q(z|x_{en}, x_{vi})$. Interestingly, many of the translations are not exact but instead capture a close interpretation of the true meaning. The MVAE tended to perform better on shorter sentences.

[9] Han Xiao, Kashif Rasul, and Roland Vollgraf. Fashion-mnist: a novel image dataset for benchmarking machine learning algorithms. *arXiv preprint arXiv:1708.07747*, 2017.

[10] Shuo Yang, Ping Luo, Chen-Change Loy, and Xiaoou Tang. From facial parts responses to face detection: A deep learning approach. In *Proceedings of the IEEE International Conference on Computer Vision*, pages 3676–3684, 2015.

| Type | Sentence |
|---|---|
| $x_{\text{en}} \sim p_{\text{data}}$ | this was one of the highest points in my life. |
| $x_{\text{vi}} \sim p(x_{\text{vi}}|z(x_{\text{en}}))$ | Đó là một gian tôi vời của cuộc đời tôi. |
| GOOGLETRANSLATE$(x_{\text{vi}})$ | It was a great time of my life. |
| $x_{\text{en}} \sim p_{\text{data}}$ | i am on this stage . |
| $x_{\text{vi}} \sim p(x_{\text{vi}}|z(x_{\text{en}}))$ | tôi đi trên sân khấu . |
| GOOGLETRANSLATE$(x_{\text{vi}})$ | me on stage . |
| $x_{\text{en}} \sim p_{\text{data}}$ | do you know what love is ? |
| $x_{\text{vi}} \sim p(x_{\text{vi}}|z(x_{\text{en}}))$ | Đó yêu của những ? |
| GOOGLETRANSLATE$(x_{\text{vi}})$ | that's love ? |
| $x_{\text{en}} \sim p_{\text{data}}$ | today i am 22 . |
| $x_{\text{vi}} \sim p(x_{\text{vi}}|z(x_{\text{en}}))$ | hãy nay tôi sẽ tuổi . |
| GOOGLETRANSLATE$(x_{\text{vi}})$ | I will be old now . |
| $x_{\text{en}} \sim p_{\text{data}}$ | so i had an idea . |
| $x_{\text{vi}} \sim p(x_{\text{vi}}|z(x_{\text{en}}))$ | tôi thế tôi có có thể vài tưởng tuyệt . |
| GOOGLETRANSLATE$(x_{\text{vi}})$ | I can have some good ideas . |
| $x_{\text{en}} \sim p_{\text{data}}$ | the project's also made a big difference in the lives of the \<unk\> . |
| $x_{\text{vi}} \sim p(x_{\text{vi}}|z(x_{\text{en}}))$ | tôi án này được ra một Điều lớn lao cuộc sống của chúng người sống chữa hưởng . |
| GOOGLETRANSLATE$(x_{\text{vi}})$ | this project is a great thing for the lives of people who live and thrive . |

Table 7: A few examples of Vietnamese MVAE translations of English sentences sampled from the empirical dataset, $p_{\text{data}}$. We use Google translate to re-translate back to English.

| Type | Sentence |
|---|---|
| $x_{\text{vi}} \sim p_{\text{data}}$ | Đó là thời điểm tuyệt vọng nhất trong cuộc đời tôi . |
| $x_{\text{en}} \sim p(x_{\text{en}}|z(x_{\text{vi}}))$ | this is the most bad of the life . |
| GOOGLETRANSLATE$(x_{\text{vi}})$ | it was the most desperate time in my life . |
| $x_{\text{vi}} \sim p_{\text{data}}$ | cảm ơn . |
| $x_{\text{en}} \sim p(x_{\text{en}}|z(x_{\text{vi}}))$ | thank . |
| GOOGLETRANSLATE$(x_{\text{vi}})$ | thank you . |
| $x_{\text{vi}} \sim p_{\text{data}}$ | trước tiên , tại sao chúng lại có ấn tượng xấu như vậy ? |
| $x_{\text{en}} \sim p(x_{\text{en}}|z(x_{\text{vi}}))$ | first of all, you do not a good job ? |
| GOOGLETRANSLATE$(x_{\text{vi}})$ | First, why are they so bad? |
| $x_{\text{vi}} \sim p_{\text{data}}$ | Ông ngoại của tôi là một người thật đáng \<unk\> phục vào thời ấy . |
| $x_{\text{en}} \sim p(x_{\text{en}}|z(x_{\text{vi}}))$ | grandfather is the best experience of me family . |
| GOOGLETRANSLATE$(x_{\text{vi}})$ | My grandfather was a worthy person at the time . |
| $x_{\text{vi}} \sim p_{\text{data}}$ | Đứa trẻ này 8 tuổi . |
| $x_{\text{en}} \sim p(x_{\text{en}}|z(x_{\text{vi}}))$ | this is man is 8 years old . |
| GOOGLETRANSLATE$(x_{\text{vi}})$ | this child is 8 years old . |

Table 8: A few examples of English MVAE translations of Vietnamese sentences sampled from the empirical dataset, $p_{\text{data}}$. We use Google translate to translate to English as a ground truth.