[Reviews · NeurIPS 2018]

Reviewer 1



This paper presents a generative approach to multimodal deep learning based on a product-of-experts (PoE) inference network. The main idea is to assume the joint distribution over all modalities factorises into a product of single-modality data-generating distributions when conditioned on the latent space, and use this to derive the structure and factorisation of the variational posterior. The proposed model shares parameters to efficiently handle any combination of missing modalities, and experiments indicate the model’s efficacy on various benchmark datasets. The idea is intuitive, the exposition is well-written and easy to follow, and the results are thorough and compelling. I have a few questions / comments, mainly about the relationship of this work with respect to previous approaches ([15] and [21] in the text). Firstly, the following line in the introduction is incorrect: “While fully-supervised deep learning approaches [15, 21] can learn to bridge modalities, generative approaches promise to capture the joint distribution across modalities and flexibly support missing data”. Both [15] and [21] are unsupervised generative approaches used to model the joint distribution, and have the machinery to deal with missing modalities. Furthermore, [21] employs a restricted Boltzmann machine, which is itself a product-of-experts model, so I would like to see a better justification for how the PoE is incorporated in the proposed approach, and how this PoE differs from [21]. Secondly, the sub-sampled training paradigm in section 2.2 is also quite similar to the denoising training approaches used in [15] and [21] (which drop out modalities randomly), so I think this could be acknowledged and the proposed approach contrasted. Lines 106-107 are also a bit unclear: is the procedure to compute the ELBO using either (1) all modalities, (2) a single modality, or (3) k randomly selected modalities; each with equal probability? I would rewrite that sentence for clarity. Lastly, two related points: (1) while the approach is benchmarked against BiVCCA, JMVAE, and CVAE, there are no comparisons some of the less recent approaches to multimodal deep learning ([15] and [21]); (2) while the experiments are comprehensive on MNIST and CelebA, these benchmarks are a little contrived in the multimodal learning case, and I believe the paper would be strengthened with a result on a more real-world multimodal problem. An additional experiment on the MIR Flickr dataset would address both of these points. Minor nitpicks: - Figure 1 caption talks about PoE, but the diagram refers to PoG. - Line 126, I would change “we’ve argued...” to a more formal tone. Ie. “we have previously argued in Section 2...” [15] Jiquan Ngiam, Aditya Khosla, Mingyu Kim, Juhan Nam, Honglak Lee, and Andrew Y Ng. Multimodal deep learning. In Proceedings of the 28th international conference on machine learning (ICML-11), pages 689–696, 2011. [21] Nitish Srivastava and Ruslan R Salakhutdinov. Multimodal learning with deep boltzmann machines. In Advances in neural information processing systems, pages 2222–2230, 2012. --AFTER REBUTTAL-- I have read the authors' response and other reviews and my overall assessment has not changed, but there is one critical aspect that should be addressed. After discussions with the AC and reviewers, we believe the claim in line 73 is not correct - the correct posterior p being factorised doesn't necessarily mean the optimal approximate posterior q when using a given family of distributions for q will also be factorised. I the argument should be changed to indicate the factorisation of p was inspiration for the form of q rather than claiming the "correct" q must be factorised, and making the argument specific to the multi-modality application as well. I have a number of other comments that I feel should be addressed before the final version: 1) The authors recognise that comparison to an RBM is interesting (given it is also a PoE) and promise a new RBM baseline in the final draft. To clarify, I think this should be the same deep architecture, trained with a shared RBM representation, rather than just a single layer RBM, to make it a fair comparison (something like [21]. Obviously the proposed approach is end-to-end and does not require layer-wise training like a Deep Boltzmann machine, an important distinction that should be emphasised. 2) The response document states that "the shared representation in [15] is not a probabilistic measure", but this does not prevent it from being a generative model as the modality-specific latent representations are stochastic and it is still modelling the data-generating distribution. As such, I think it would be incorrect to state (as in the original draft), that [15] and [21] are not generative models. I believe this should be rectified for both, not just for [21]. 3) The additional text translation task is compelling - but I assume that the baseline performances will be reported on this task as well? 4) My comment on the sub-sampling training of the model was, I believe, only partially addressed. In particular, I still think the section detailing this process could be written more clearly, in terms of what data is used and how this relates to previous "denoising training" procedures (eg. [15] and [21]), which I believe are effectively sub-sampling in any case. All in all, I think this would be a good addition to the conference.

Reviewer 2



Summary: The main contribution of this paper is the proposed multimodal variational autoencoder (MVAE) for learning a joint distribution under weak supervision. The key novelty of MVAE is the use of a product-of-experts inference network and a sub-sampled training paradigm to solve the multi-modal inference problem. The authors first present experiments on several two-modalities datasets, and further apply MVAE to problems with more than two modalities. Strengths: (1) Clarity: The paper is clearly presented and easy to follow. I enjoyed reading the paper. (2) Originality: The proposed model is simple but also novel. Firstly, it seems novel to use a product-of-experts inference network to do the posterior inference of latent variables. Each individual expert shares its statistical strength to enhance each other, and can deal with missing modalities naturally, thus suitable for weakly-supervised learning. Second, the proposed sub-sampled training paradigm is also intuitive and interesting. (3) Significance: I think the paper proposes a nice approach for multimodel learning, especially when dealing with multiple modalities, instead of only two. (4) Quality: The quality of this paper is also good. Methods are clearly described, experiments are carefully designed, with more details also included in the supplement. However, I also have one major concern listed below. Weaknesses: (1) Quality: I think the experiments in Section 5.1 are not carefully designed. As we can see from the title, this paper cares about weakly-supervised learning, and also since the proposed approach will be especially powerful when dealing with missing modalities, I think the experiments in Section 5.1 need more care. Specifically, in Figure 3, the authors shows the performance of MVAE, using accuracy as the evaluation metric, which is good. However, no comparison with other methods are provided. The numbers in Table 2 & 3 are both for fully supervised learning. I'm wondering how the model performs when compared with other methods, like VAE, BiVCCA, JMVAE etc. in the weakly-supervised setting. And especially, how it performs compared with the state-of-the-art? This set of results should be provided to demonstrate the power of the model for weakly-supervised/semi-supervised learning. We need not only a curve like in Figure 3, but also detailed numbers and compare with others. For example, for MNIST classification, it should be easy to find a lot of baseline results in the literature. Missing reference: Variational Autoencoder for Deep Learning of Images, Labels and Captions, NIPS 2016. I recommend the authors also citing the above paper. This NIPS 2016 paper shares the same graphical model as illustrated in Figure 1(a). Further, the authors considered image-label modalities, and image-caption modalities, and presented results on using VAE for semi-supervised image classification and image captioning, which I think is related to the topic in this submission. Overall, I vote accept for this paper, and the paper has the potential to be a better one by providing more comprehensive results on weakly-supervised learning. Minor issues: (1) Line 73: see the equation in the rightmost of the line, "n" => "N". (2) Table 3: "marginal, probabilities" => "marginal probabilities" ---- AFTER REBUTTAL ---- I have read the authors' response and other reviewers' comments, and decide to keep my original score. As agreed by all the reviewers, this work has some novelty inside, and well-presented. The added new experiment on text translation seems also interesting. The authors claimed that they will add detailed numbers and additional baselines in the weak supervision section. This addresses my concern, and therefore, in summary, I prefer to voting for acceptance of the paper. During the discussion, we noticed that although the derivation of Eqn. (3) has no problem, the claim in Line 73 "This suggests that the correct q is a product of experts" is not precise. Instead of considering x_1, ... x_N as N modalities, if x_1, ..., x_N are N pixels, and are conditionally independent given z, which is the assumption of many VAE models, then Line 73 implies that the most correct q is a PoE which uses one inference network per dimension without sacrificing expressive power. This seems strange, and we think the fact that the factorized true posterior is a PoE does not necessarily mean the variational posterior should also be a PoE, so we recommend the authors changing the claim in Line 73 to something like "the usage of a PoE inference network is more like a theoretically motivated heuristic".

Reviewer 3



This paper proposes a generative model for multi-modal data, i.e. different aspects of the same underlying "concept" (image and its caption). The modalities are considered conditional independent given a latent variable z, trained with a VAE-type loss by approximating the posterior distribution p(z|x_1, ..., x_N), where x is a modality. The novelty of the paper is to show that this posterior can be a PoE (product-of-expert) distribution which have a closed form solution in case of multivariate Gaussians and, as the authors show, is a reasonable approximation of the true posterior (itself a product of posteriors corresponding to each modality). The use of the PoE make the model scale to multiple modalities, without having to train a posterior for each combination of observed modalities. The authors show that their proposed posterior approximation is robust when only some of the examples contained all the modalities in the joint (weak supervision). Pros: - The paper reads very well and the math is sound. The authors do a good job in explaining the model and the contributions. - The proposed solution to make approximate posterior scalable is simple and seems as least as effective as other models. Cons: - I feel that the novelty of the paper (the approximate posterior q(z|X) is a product of experts of the posteriors for each modality \prod_i q(z|x_i)) is rather small. - From Table 2, I cannot see a clear difference between the proposed approaches. The reported LL scores are very close to each other. - It would have been interesting to have experiments in more realistic settings that could better highlight the usefulness / attractiveness of the general problem, i.e. images and captions in different languages , maybe ? -- After rebuttal I thank the authors for the additional effort in making a strong rebuttal. This has contributed to reinforce the contribution as well as to give a nice example on how this method could be applicable in more realistic setting. Therefore, I am willing to increase the overall score.